# Improving the Estimation of Canopy Fluorescence Escape Probability in the Near-Infrared Band by Accounting for Soil Reflectance

Mengjia Qi [1,2,3], Xinjie Liu [1,2,*], Shanshan Du [1,2], Linlin Guan [1,2], Ruonan Chen [1,2,3] and Liangyun Liu [1,2,3]

1   Key Laboratory of Digital Earth Science, Aerospace Information Research Institute, Chinese Academy of Sciences, Beijing 100094, China; qimengjia21@mails.ucas.ac.cn (M.Q.); duss@radi.ac.cn (S.D.); guanll@aircas.ac.cn (L.G.); chenruonan19@mails.ucas.ac.cn (R.C.); liuly@radi.ac.cn (L.L.)
2   International Research Center of Big Data for Sustainable Development Goals, Beijing 100094, China
3   College of Resources and Environment, University of Chinese Academy of Sciences, Beijing 100049, China
*   Correspondence: liuxj@radi.ac.cn

**Abstract:** Solar-induced chlorophyll fluorescence (SIF) has been found to be a useful indicator of vegetation's gross primary productivity (GPP). However, the directional SIF observations obtained from a canopy only represent a portion of the total fluorescence emitted by all the leaf photosystems because of scattering and reabsorption effects inside the leaves and canopy. Hence, it is crucial to downscale the SIF from canopy level to leaf level by modeling fluorescence escape probability ($f_{esc}$) for improved comprehension of the relationship between SIF and GPP. Most methods for estimating $f_{esc}$ rely on the assumption of a "black soil background," ignoring soil reflectance and the effect of scattering between soils and leaves, which creates significant uncertainties for sparse canopies. In this study, we added a correction factor considering soil reflectance, which was modeled using the Gaussian process regression algorithm, to the semi-empirical NIRv/FAPAR model and obtained the improved $f_{esc}$ model accounting for soil reflectance (called the $f_{esc\_GPR-SR}$ model), which is suitable for near-infrared SIF downscaling. The evaluation results using two simulation datasets from the Soil–Canopy–Observation of Photosynthesis and the Energy Balance (SCOPE) model and the Discrete Anisotropic Radiative Transfer (DART) model showed that the $f_{esc\_GPR-SR}$ model outperformed the NIRv/FAPAR model, especially for sparse vegetation, with higher accuracy for estimating $f_{esc}$ ($R^2$ = 0.954 and RMSE = 0.012 for SCOPE simulations; $R^2$ = 0.982 and RMSE = 0.026 for DART simulations) compared with the NIRv/FAPAR model ($R^2$ = 0.866 and RMSE = 0.100 for SCOPE simulations; $R^2$ = 0.984 and RMSE = 0.070 for DART simulations). The evaluation results using in situ observation data from multi-species canopies also suggested that the leaf-level SIF calculated by the $f_{esc\_GPR-SR}$ model tracked better with photosynthetic active radiation absorbed by green components (APAR$_{green}$) for sparse vegetation ($R^2$ = 0.937, RMSE = 0.656 mW/m$^2$/nm) compared with the NIRv/FAPAR model ($R^2$ = 0.921, RMSE = 0.904 mW/m$^2$/nm). The leaf-level SIF calculated by the $f_{esc\_GPR-SR}$ model was less sensitive to observation angles and differences in canopy structure among multiple species. These results emphasize the significance of accounting for soil reflectance in the estimation of $f_{esc}$ and demonstrate that the $f_{esc\_GPR-SR}$ model can contribute to further exploring the physiological mechanism between SIF and GPP.

**Keywords:** solar-induced chlorophyll fluorescence (SIF); fluorescence escape probability; soil reflectance; Gaussian process regression; downscaling

## 1. Introduction

Solar-induced chlorophyll fluorescence (SIF) is a phenomenon in which chlorophyll molecules in plants emit light in response to natural light. It is a byproduct of the light reaction process of photosynthesis [1,2] and has been proven to serve as a proxy for gross primary productivity (GPP) in numerous studies [2–6]. The light energy absorbed by

photosynthetic pigments can be dissipated through three main pathways: photosynthesis, heat dissipation, and fluorescence [2,7]. SIF changes in sync with photochemical quenching and competes with heat dissipation under natural light conditions without stress, and it is closely tied to photosynthesis in the basic physiological and biochemical processes of plants [8,9]. As such, SIF provides a more accurate measurement of photosynthetic ecological changes and has a more direct relationship with GPP compared with vegetation indices based on reflectance [10,11].

There are currently many algorithms that can successfully retrieve SIF from ground-based and satellite-based observations [8,9,12–14]. However, the fluorescence photons observed at the canopy level only represent a fraction of the total SIF emission due to the effect of the radiative transfer process. After fluorescence photons are emitted by photosystems, they are absorbed and scattered by the leaf and canopy components and then escape to the canopy, where they are detected by sensors [2]. The red SIF is primarily affected by chlorophyll absorption within leaves, whereas the near-infrared SIF is primarily impacted by the scattering effect within the canopy [15]. The scattering and reabsorption effects result in different SIF–GPP relationships due to variations in canopy structure, such as leaf area, leaf orientation, and leaf clumping [16]. Furthermore, multiple scatterings inside the canopy during SIF transmission make the SIF observed from different angles vary, which demonstrates that canopy SIF is directional and not isotropic [17]. As a result, the canopy-level SIF differs from the leaf-level SIF and cannot be utilized to directly quantify changes in plant physiology [5,18,19].

To reduce the influence of the canopy structure and directional effect on the SIF–GPP relationship, it is necessary to downscale the SIF from canopy level to leaf level and obtain the leaf-level SIF which has a closer physiological coupling relationship with GPP. The fluorescence escape probability ($f_{esc}$) represents the probability that the fluorescence emitted by photosystems escapes from the canopy and is an important bridge connecting the canopy SIF ($SIF_{canopy}$) and the total SIF emitted by photosystems ($SIF_{total}$). Their relationship can be expressed as follows [20–25]:

$$SIF_{canopy} = SIF_{total} \times f_{esc} = PAR \times FAPAR \times \Phi_{SIF} \times f_{esc} \qquad (1)$$

where PAR is the incident photosynthetically active radiation; FAPAR is the fraction of photosynthetically active radiation absorbed by vegetation; $\Phi_{SIF}$ is the fluorescence quantum yield. The weak absorption effect of leaves in the near-infrared band means that the fluorescence escape probability from photosystems to leaves can be approximated by the leaf albedo with a value around 1 [26], and the $f_{esc}$ in Equation (1) can be approximated as the fluorescence escape probability from leaves to the canopy and its unit is $sr^{-1}$.

In recent years, a number of studies have concentrated on estimating $f_{esc}$, in other words, downscaling SIF from the canopy to the leaf level; however, a common problem in most of these approaches is that they are based on spectrally invariant properties and the assumption of a "black soil background," assuming that the soil is a black body with a reflectance of 0, which absorbs all external radiation signals, regardless of the reflective property of the soil and the effects of multiple scatterings among soils and leaves. Yang et al. [27] proposed that $f_{esc}$ could be accurately estimated using near-infrared reflectance, canopy interception ($i_0$), and leaf albedo in the near-infrared band ($\omega_N$), assuming no soil reflectance ($f_{esc} = Ref_{NIR}/i_0 \cdot \omega_N$), which laid the foundation for the study of $f_{esc}$ estimation. Liu et al. [15] used this model to estimate $f_{esc}$ from canopy reflectance based on random forest regression, effectively avoiding the difficulties of estimating $i_0$ and $\omega_N$ in the model. Zeng et al. [28] further simplified the parameters in the model by developing the near-infrared reflectance of vegetation index (NIRv) to isolate the contribution of soil and vegetation to canopy reflectance and using FAPAR to approximate $i_0$ in the denominator. Similarly, Liu et al. [26] derived fluorescence escape probability formulas for the red and near-infrared bands based on reflectance and FAPAR, respectively, and applied them to correct long-term ground observation data of SIF. Yang et al. [29] later showed that it was challenging to estimate FAPAR and $f_{esc}$ in the near-infrared band separately using canopy

reflectance alone; however, the product of the two could be well replaced with a new vegetation index, the fluorescence-corrected vegetation index (FCVI), which was shown to be in good agreement with spectrally invariant properties.

Among the above-mentioned SIF-downscaling methods, the semi-empirical model proposed by Zeng et al. [28] is simple and accurate, facilitating the use of current in situ or remotely sensed NIRv and FAPAR datasets for $f_{esc}$ estimation. As a result, Zeng et al.'s model has been widely used and can be expressed as:

$$f_{esc} \approx \frac{NIRv}{i_0 \times \omega_N} = \frac{NDVI \times Ref_{NIR}}{i_0 \times \omega_N} \approx \frac{NIRv}{FAPAR} \qquad (2)$$

where $f_{esc}$ is a dimensionless quantity, different from $f_{esc}$ (unit: $sr^{-1}$) in Equation (1). For ease of understanding, the $f_{esc}$ values in the following sections are all expressed as dimensionless quantities. The canopy interception, $i_0$, is approximated by FAPAR. The leaf single scattering albedo in the near-infrared band (leaf reflectance + transmittance), $\omega_N$, is taken to be constant and equal to 1. Although the use of NIRv in the model is intended to eliminate the effect of soil reflectance, in practice, NIRv as a pure vegetation signal is controversial because soil reflectance impacts the calculation of the normalized difference vegetation index (NDVI). Additionally, the approximation of $i_0$ with FAPAR relies on the assumption that soil reflectance is 0 [29]. Hence, the NIRv/FAPAR model proposed by Zeng et al. does not completely eliminate the impact of soil reflectance, which remains a key source of uncertainty in $f_{esc}$ estimation, especially for sparse vegetation. The influence of soil reflectance on $f_{esc}$ estimation can be seen in two aspects: its impact on the calculation of pure vegetation reflectance and on the scattering process between the canopy and soil.

To thoroughly correct the effect of soil reflectance, we aim to add a correction factor that includes soil reflectance to the simple NIRv/FAPAR model to increase the precision of $f_{esc}$ and leaf-level SIF estimation in the near-infrared band. First, we used the Soil–Canopy–Observation of Photosynthesis and the Energy Balance (SCOPE) model to simulate the training dataset. Then, we employed the Gaussian process regression (GPR) algorithm to model the correction factor, obtaining the improved $f_{esc}$ modeling method accounting for soil reflectance ($f_{esc\_GPR-SR}$ model). Finally, we assessed the performance of the $f_{esc\_GPR-SR}$ model using simulated data, including the SCOPE model and the Discrete Anisotropic Radiative Transfer (DART) model, combined with field-measured data.

## 2. Materials and Methods

### 2.1. Simulated Datasets

#### 2.1.1. SCOPE Model Simulations

The SCOPE model [30] is a one-dimensional radiative transfer model that simulates the interaction between radiative transport, microscopic meteorological processes, spectral reflectance, SIF, and photosynthetic and hydrothermal fluxes in leaves and canopies. We used SCOPE v1.73 to simulate canopy-level SIF, leaf-level SIF, and canopy directional reflectance.

Leaf absorption is primarily influenced by chlorophyll concentration [31], while canopy scattering is primarily influenced by Sun–target–viewing geometries and structural parameters such as leaf area index (LAI) and leaf inclination distribution function (LIDF) [32]. We parameterized the model with a range of leaf chlorophyll a and b content ($C_{ab}$), LAI, and five typical LIDFs (excluding Erectophile, which is uncommon) to cover the most common vegetation states. Since only the soil reflectance at 780 nm is used for model training, it has nothing to do with the shape of the soil spectra. It is only necessary to ensure that the soil spectra input of the SCOPE model covers the common soil reflectance range in the near-infrared band [33] (Figure 1). We selected small LAI values densely to represent sparse vegetation conditions. The details of the input variables for the SCOPE model are listed in Table 1. In total, 81,000 distinct samples were generated. To improve model training efficiency and reduce the processing time, we randomly selected 9000 samples for training (1500 simulated samples per soil spectral curve, 1500 × 6 = 9000), and another 9000 samples were selected for validation.

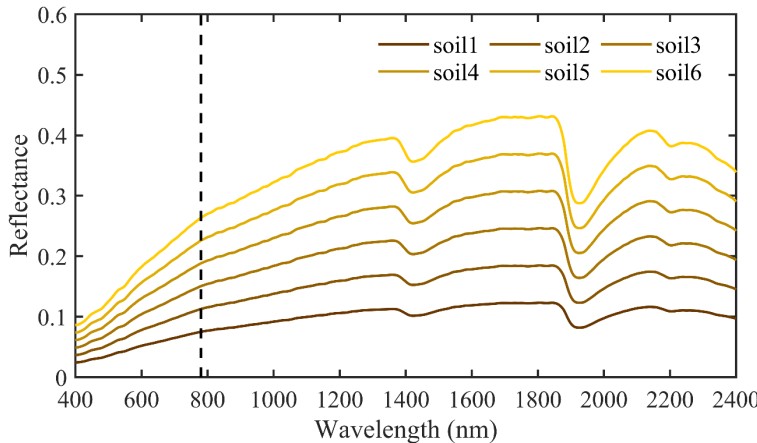

**Figure 1.** Soil spectral curves used in the SCOPE model. The black dashed line shows the wavelength position at 780 nm.

**Table 1.** Main input variables of the SCOPE model.

| Variables | Definition | Values | Unit |
| --- | --- | --- | --- |
| Cab | Leaf chlorophyll a and b content | 20, 40, 60, 80 | $\mu g/cm^2$ |
| LAI | Leaf area index | 0.25, 0.5, 0.75, 1, 1.5, 2, 3, 5, 7 | $m^2/m^2$ |
| $LIDF_a$ | Leaf inclination parameter | 1, 0, 0, −0.35, 0 | − |
| $LIDF_b$ | Bimodality parameter | 0, −1, 1, −0.15, 0 | − |
| SZA | Solar zenith angle | 20, 30, 40, 50, 60 | Degree |
| VZA | Viewing zenith angle | 0, 15, 30, 45, 60 | Degree |
| RAA | Relative azimuth angle | 0, 90, 180 | Degree |
| Soil spectra | Soil reflectance | Six soil spectra | − |

### 2.1.2. DART Model Simulations

The DART model [34] is a three-dimensional radiative transfer model that simulates the propagation of radiation across the whole optical domain, from the visible to the thermal infrared region, for natural landscapes such as forests, grasslands, and farmland, as well as for urban landscapes with topography and atmosphere. Recently, the FLUSPECT model was integrated into the DART model to simulate the radiative transfer of SIF within 3-D canopies [35]. In this study, the accuracy of the $f_{esc}$ was verified using DART v5.6.7, which simulated the SIF at both canopy and leaf levels, as well as the directional reflectance for fifty various viewing angles for maize canopies. The input variables for the DART model are listed in Table 2. To simulate sparse maize canopies, the value of LAI was set to 2. The soil spectrum from the DART model database was used (Figure S1). The specific simulated 3-D maize canopy scene is shown in Figure S2. Figure 2 shows the simulated multi-angle canopy SIF results in the near-infrared band (760 nm).

**Table 2.** Main input variables of the DART model.

| Variables | Definition | Values | Unit |
| --- | --- | --- | --- |
| Vegetation type | Vegetation type | Maize | − |
| N | Structure coefficient | 1.5 | − |
| $C_{ab}$ | Leaf chlorophyll a and b content | 58 | $\mu g/cm^2$ |
| Yield PSI | Fluorescence quantum yield for photosystem I | 0.002 | − |
| Yield PSII | Fluorescence quantum yield for photosystem II | 0.008 | − |
| LAI | Leaf area index | 2 | $m^2/m^2$ |

**Table 2.** *Cont.*

| Variables | Definition | Values | Unit |
|---|---|---|---|
| Canopy height | Canopy height | 1.5 | m |
| Soil spectra | Soil reflectance | loam_gravelly_brown_dark | – |
| SZA | Solar zenith angle | 30.9303 | Degree |
| SAA | Solar azimuth angle | 249.1069 | Degree |
| VZA | Viewing zenith angle | 0–90 | Degree |
| VAA | Viewing azimuth angle | 0–360 | Degree |

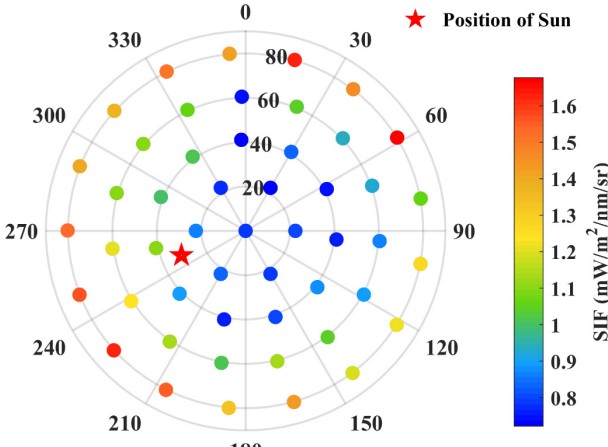

**Figure 2.** Multi-angle canopy-level SIF results of maize canopies in the near-infrared band (760 nm) using the DART model. The angular coordinate and diameter coordinate represent the viewing azimuth angles (VAAs, 0–360°) and the viewing zenith angles (VZAs, 0–90°), respectively. The red pentagram indicates the position of the sun (SZA = 30.9303°, SAA = 249.1069°).

*2.2. Field Dataset*

2.2.1. Field-Measured Dataset

The in situ spectral data were collected from four field experiments carried out at three locations in 2016. The data were employed to assess the performance of the improved $f_{esc}$ estimation model in canopies with varying structures. The first and second spectral measurements of winter wheat were taken at the Xiaotangshan Farm (XTS) in Beijing on 8–9 April, 18 April, and 8 December, when the winter wheat was at the stages of jointing, booting, and tillering, respectively. The third experiment was conducted on 18 December at the Nanbin Farm (NBF) in Sanya and included vegetables and crops such as sweet potato, cotton, pumpkin, and maize. The fourth spectral measurement was taken on 18 December at the Sanya Remote Sensing Satellite Data Receiving Station (SYS) and included gold coin grass. The details of these four ground measurements are listed in Table 3. According to the visual judgment of canopies in Figure 3, the LIDF type in XTS was spherical and, in NBF and SYS, it was planophile. All the spectral measurements were performed using a custom-made Ocean Optics QE Pro spectrometer (Ocean Optic, Inc., Dunedin, FL, USA).

**Table 3.** Detailed information of the four ground measurements.

| Sites | Xiaotangshan Farm | Xiaotangshan Farm | Nanbin Farm | Sanya Station |
|---|---|---|---|---|
| Location | 40°11′N 116°27′E | 40°11′N 116°27′E | 18°22′N 109°10′E | 18°18′N 109°18′E |
| Dates in 2016 | 8, 9, 18 April | 8 December | 18 December | 18 December |
| Species | Winter wheat | Winter wheat | Vegetables and crops | Gold coin grass |
| Fractional vegetation cover (FVC) | 0.72–0.79 | 0.21–0.63 | 0.28–0.91 | 0.67 |
| Soil reflectance in NIR band * | 0.12–0.17 | 0.13–0.16 | 0.09–0.16 | 0.11–0.13 |

* Soil reflectance in NIR band included that of litters.

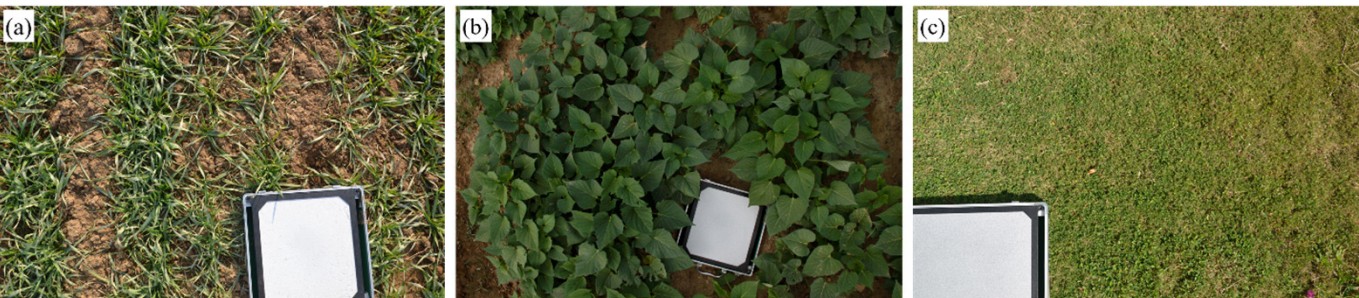

**Figure 3.** Photographs of canopies at (**a**) XTS in December (winter wheat), (**b**) NBF (sweet potato), and (**c**) SYS (gold coin grass). Only one representative photo is shown at each site here.

### 2.2.2. SIF Retrieval

The three-band fluorescence line discriminator (3FLD) method [36] is simple and reliable for data with 0.3 nm spectral resolution, according to Damm et al. [37] and Liu et al. [38]. This method was selected in this study for retrieving SIF. The formula for the 3FLD method is expressed as follows:

$$\text{SIF}_{\text{canopy}} = \frac{(I_{\text{left}} \times w_{\text{left}} + I_{\text{right}} \times w_{\text{right}}) \times L_{\text{in}} - I_{\text{in}} \times (L_{\text{left}} \times w_{\text{left}} + L_{\text{right}} \times w_{\text{right}})}{(I_{\text{left}} \times w_{\text{left}} + I_{\text{right}} \times w_{\text{right}}) - I_{\text{in}}} \quad (3)$$

where I is the incident solar radiance reaching the top of the canopy; L is the entire upwelling radiance; and the weight w, which is inversely proportional to the distance between the left- and right-side bands and the inner band, is used to average I and L outside the absorption band. The subscripts in, left, and right denote the bands that are within, to the left of, and to the right of the absorption band, respectively. As noted by Liu et al. [38] and Liu et al. [22], the wavelengths corresponding to the left, inner, and right shoulder of the absorption feature for the $O_2$-A band are 752.92 nm, 760.72 nm, and 768.87 nm, respectively.

### 2.2.3. Estimation of APAR$_{\text{green}}$

Because only the green components of the canopy can carry out photosynthesis, the FAPAR for the entire canopy can be divided into the photosynthetic active green components (FAPAR$_{\text{green}}$) and the non-photosynthetic active components (FAPAR$_{\text{non-green}}$) [39]. Liu et al. [40] proposed an in situ measurement approach of FAPAR$_{\text{green}}$ in the low vegetation canopy using a digital camera and reference panel, which has been proven to be effective. Using this method, FAPAR$_{\text{green}}$ can be calculated as follows:

$$\text{FAPAR}_{\text{green}} = \frac{\text{PAR}_i - \text{PAR}_r - (\text{APAR}_{\text{exp\_b}} + \text{APAR}_{\text{cov\_b}})}{\text{PAR}_i} \quad (4)$$

where PAR$_i$ and PAR$_r$ are the incident and reflected PAR calculated from the DN values of digital photographs. APAR$_{\text{exp\_b}}$ and APAR$_{\text{cov\_b}}$ are the PAR absorbed by the exposed background and the vegetation-covered background, respectively. Consequently, photosynthetically active radiation absorbed by green components (APAR$_{\text{green}}$) can be further deduced as:

$$\text{APAR}_{\text{green}} = \text{PAR} \times \text{FAPAR}_{\text{green}} \quad (5)$$

### 2.3. Correction Factor Accounting for Soil Reflectance

The NIRv/FAPAR model, proposed by Zeng et al. [28] (Equation (2)), is currently a popular method for downscaling SIF in the near-infrared band. However, this method is limited by the fact that it does not explicitly consider the impact of soil reflectance, which can result in errors in $f_{\text{esc}}$ estimation, particularly for sparse vegetation. Therefore, a correction factor is necessary to optimize the estimation of $f_{\text{esc}}$ in Zeng et al. [28] to reduce

the impact of soil reflectance. The contribution of soil background to $f_{esc}$ estimation is not only related to its reflectance but also to vegetation coverage, which can be represented by the vegetation index. Hence, the correction factor can be expressed as a function of soil reflectance in the near-infrared band and vegetation index—$f$ (*Ref_soil*, *VI*). In this study, four vegetation indices (NDVI, simple ratio (SR), two-band enhanced vegetation index without the blue band (EVI2), and perpendicular vegetation index (PVI)) were selected to characterize vegetation structure (Table 4). The correction factor was added to the widely used NIRv/FAPAR model by multiplying and adding functions, resulting in two improved models:

$$f_{esc} = \frac{NIRv}{FAPAR} \times f_1(Ref_{soil}, VI) \tag{6}$$

$$f_{esc} = \frac{NIRv}{FAPAR} + f_2(Ref_{soil}, VI) \tag{7}$$

where $f_1$ (*Ref_soil*, *VI*) and $f_2$ (*Ref_soil*, *VI*) are the two correction factors for multiplicative correction and additive correction, respectively.

**Table 4.** VIs used for the calculation of the correction factor ($R_{780}$, $R_{710}$, and $R_{678}$ represent the reflectance at 780 nm, 710 nm, and 678 nm, respectively).

| VIs | References |
|---|---|
| $NDVI = (R_{780} - R_{678})/(R_{780} + R_{678})$ | [41] |
| $SR = R_{780}/R_{678}$ | [42] |
| $EVI2 = \frac{2.5 \times (R_{780} - R_{710})}{R_{780} + 2.4 \times R_{678} + 1}$ | [43] |
| $PVI = \sqrt{(R_{780}^{soil} - R_{780}^{veg})^2 + (R_{678}^{soil} - R_{678}^{veg})^2}$ | [44] |

To improve the robustness of $f_{esc}$ estimation, a machine learning method was employed to estimate the correction factor $f$ (*Ref_soil*, *VI*). A number of machine learning algorithms were tested for model training, including decision tree, random forest, support vector machine, and Gaussian process regression (GPR), and it was found that the exponential Gaussian process regression [45] was the best training model in this study (Figure S3), which can quantitatively determine the prediction uncertainty in a systematic manner. The model was trained using the 5-fold cross-validation method. The detailed GPR model parameters can be found in Table S1. According to Equations (6) and (7), the input parameters of the two machine learning models were *Ref_soil* and *VI*, and the outputs were $f_{esc}/\frac{NIRv}{FAPAR}$ and $f_{esc} - \frac{NIRv}{FAPAR}$, respectively. The Gaussian regression algorithm was used to train two different correction factors, $f_1$ (*Ref_soil*, *VI*) and $f_2$ (*Ref_soil*, *VI*), in the models. All input and output parameters can be obtained from the SCOPE simulations.

## 3. Results

### 3.1. Performance of Different Correction Factors for $f_{esc}$ Estimation

As discussed in Section 2.3, vegetation indices should be included in the correction factor of the model. In this study, NDVI, SR, EVI2, and PVI were evaluated. The determination coefficient ($R^2$), root mean square error (RMSE), and mean absolute error (MAE) were used to evaluate the models' performance. As depicted in Figure 4, model $f_{esc}/\frac{NIRv}{FAPAR}$ performed significantly better than model $f_{esc} - \frac{NIRv}{FAPAR}$. However, changing the vegetation indices had little effect on the $R^2$ values. By comparing other performance evaluation indices (Table 5), we found that NDVI performed best under all circumstances, including sparse vegetation. NDVI is the most extensively used of over 40 vegetation indices [46] and has been proven as a reliable representation of LAI and FVC [47,48]. Hence, NDVI was finally chosen as the vegetation index in the correction factor.

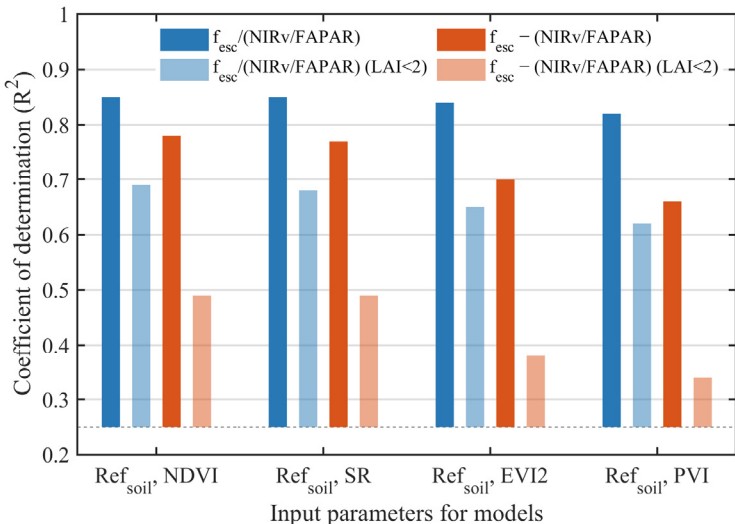

**Figure 4.** Comparison of the determination coefficients $R^2$ of exponential GPR models with different input and output parameter combinations. The horizontal coordinate represents the inputs of the model, and the color of the bars represents the output of the model. Light blue and light orange bars indicate the results with LAI < 2.

**Table 5.** Statistics of performance evaluation indices of the exponential GPR method with different input parameters.

| Inputs for Models | $R^2$ | RMSE | MAE | $R^2$ (LAI < 2) | RMSE (LAI < 2) | MAE (LAI < 2) |
|---|---|---|---|---|---|---|
| $Ref_{soil}$, NDVI | 0.85 | 0.0391 | 0.0311 | 0.69 | 0.0438 | 0.0348 |
| $Ref_{soil}$, SR | 0.85 | 0.0396 | 0.0315 | 0.68 | 0.0441 | 0.0350 |
| $Ref_{soil}$, EVI2 | 0.84 | 0.0403 | 0.0313 | 0.65 | 0.0467 | 0.0372 |
| $Ref_{soil}$, PVI | 0.82 | 0.0433 | 0.0334 | 0.62 | 0.0484 | 0.0385 |

When the inputs of the model were the combination of NDVI and $Ref_{soil}$ and the output was $f_{esc}/\frac{NIRv}{FAPAR}$, the model performance $R^2$ reached 0.850, indicating that $f_1$ ($Ref_{soil}$, *NDVI*) can explain 85% of the remaining part of $f_{esc}$ from the NIRv/FAPAR model. Even for sparse canopies (LAI < 2), $R^2$ reached 0.690. Therefore, the improved $f_{esc}$ model in the near-infrared band accounting for soil reflectance (referred to as the $f_{esc\_GPR-SR}$ model) is defined as follows:

$$f_{esc} = \frac{NIRv}{FAPAR} \times f_1(Ref_{soil}, NDVI) \tag{8}$$

### 3.2. Evaluation of the $f_{esc\_GPR-SR}$ Model Using Simulated Data

3.2.1. Validation of the $f_{esc\_GPR-SR}$ Model Using SCOPE Simulations

The performance for estimating $f_{esc}$ of the $f_{esc\_GPR-SR}$ model was first assessed using the SCOPE simulations. The comparison of $f_{esc}$ calculated by the NIRv/FAPAR model and the $f_{esc\_GPR-SR}$ model with that simulated by the SCOPE model is shown in Figure 5, and it is evident that the $f_{esc}$ values estimated by the NIRv/FAPAR model are significantly underestimated (data points are below the 1:1 line), especially for canopies with low LAI values. Figure 6 clearly illustrates that as the LAI value increases, the underestimation effect of $f_{esc}$ calculated by the NIRv/FAPAR model decreases. The $f_{esc}$ estimated by the $f_{esc\_GPR-SR}$ model is in close agreement with the true $f_{esc}$ values under all LAI values, which shows that the $f_{esc\_GPR-SR}$ model can effectively correct the underestimation of $f_{esc}$, especially for sparse vegetation. The $R^2$ value of the relationship between $f_{esc}$ estimated by the $f_{esc\_GPR-SR}$ model and the true $f_{esc}$ values improves from 0.730 to 0.949, and the RMSE reduces from 0.081 to 0.015, compared with the NIRv/FAPAR model. The improvement of the $f_{esc\_GPR-SR}$ model is also significant in the case of sparse vegetation, with $R^2$ increasing from 0.866 to

0.954 and RMSE decreasing from 0.100 to 0.012. The results demonstrate that the $f_{esc\_GPR-SR}$ model provides a better estimation of $f_{esc}$ in the near-infrared band and has high estimation accuracy for sparse vegetation.

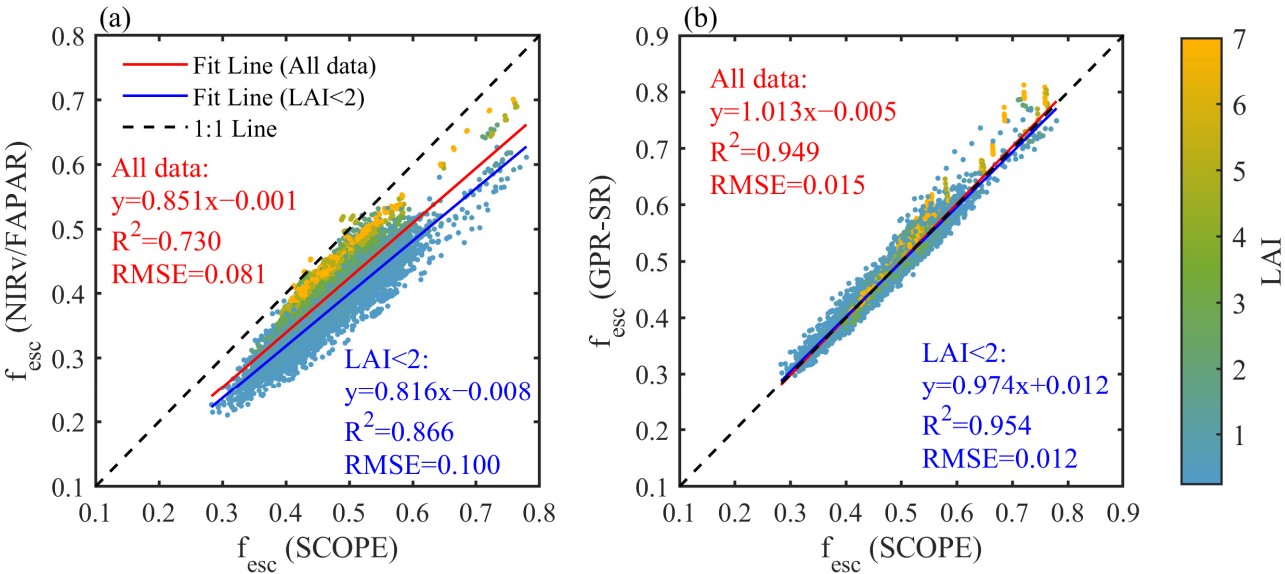

**Figure 5.** Comparison of $f_{esc}$ in the near-infrared band (760 nm) estimated by (**a**) the NIRv/FAPAR model ($f_{esc}$ (NIRv/FAPAR)) and (**b**) the $f_{esc\_GPR-SR}$ model ($f_{esc}$ (GPR-SR)) with the $f_{esc}$ simulated by the SCOPE model. $R^2$ is the correlation coefficient of linear regression, and RMSE is the root mean square error between $f_{esc}$ (SCOPE) and the $f_{esc}$ values calculated by $f_{esc}$ estimation models.

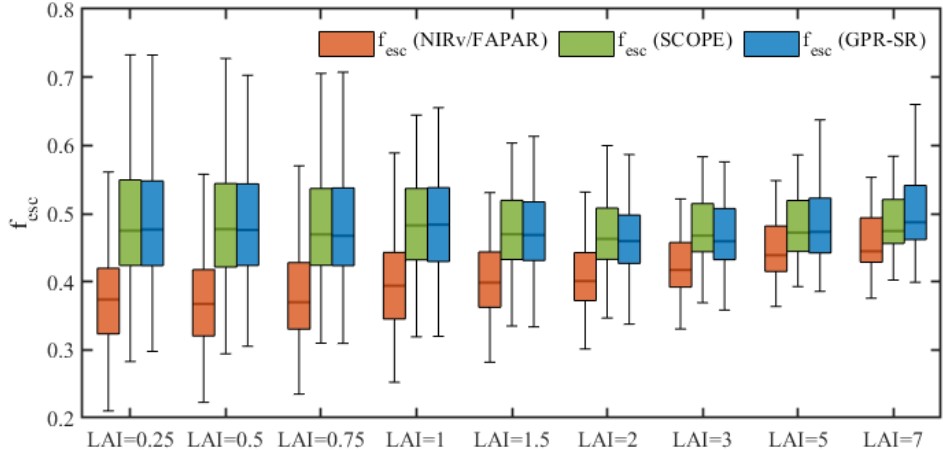

**Figure 6.** Comparison of $f_{esc}$ in the near-infrared band (760 nm) calculated by the NIRv/FAPAR model ($f_{esc}$ (NIRv/FAPAR), orange boxes), the SCOPE model ($f_{esc}$ (SCOPE), green boxes), and the $f_{esc\_GPR-SR}$ model ($f_{esc}$ (GPR-SR), blue boxes) under different LAI values.

Furthermore, we calculated the coefficient of variation (CV) of the canopy SIF simulated by the SCOPE model and the leaf SIF estimated by the $f_{esc\_GPR-SR}$ model and NIRv/FAPAR model for each set of different LAI values with only the VZA changing and the remaining parameters fixed (Figure 7). The smaller the CV value, the less the SIF is affected by the directional effect. The CV of canopy SIF for different LAI values is much larger than that of leaf SIF due to the anisotropic scattering within the canopy. Apparently, compared with the NIRv/FAPAR model, the CV of leaf SIF estimated by the $f_{esc\_GPR-SR}$ model is smaller, and the gap between the results of the two models is more obvious for sparse vegetation. The result suggests that our $f_{esc\_GPR-SR}$ model can reduce the directional effect of canopy SIF more effectively compared with the NIRv/FAPAR model.

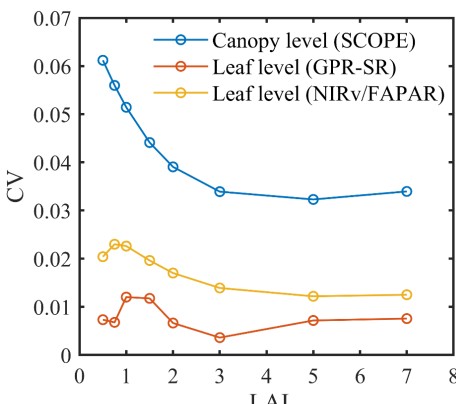

**Figure 7.** The coefficients of variation (CVs) of the canopy SIF simulated by the SCOPE model and the leaf SIF estimated by the $f_{esc\_GPR-SR}$ model and NIRv/FAPAR model for each set of different LAI values with only the VZA changing and the remaining parameters fixed (LIDF is plagiophile, $C_{ab}$ = 60 μg/cm², SZA = 30°, RAA = 180°, and soil reflectance at 780 nm is 0.1508). The amount of data for different LAI values is the same.

### 3.2.2. Validation of the $f_{esc\_GPR-SR}$ Model Using DART Simulations

The reliability of the $f_{esc\_GPR-SR}$ model was further assessed using leaf-level and canopy-level SIF of maize plants simulated by the DART model under sparse vegetation conditions (LAI = 2). The comparison of SIF at canopy and leaf levels is shown in Figure 8. For ease of comparison, the unit of leaf-level SIF was converted from mW/m²/nm to mW/m²/nm/sr. The leaf-level SIF estimated by the $f_{esc\_GPR-SR}$ model (the mean value is 1.818 mW/m²/nm/sr) is found to be closer to the SIF simulated by the DART model ($SIF_{leaf}^{DART}$ = 1.813 mW/m²/nm/sr). The interquartile range (IQR) of leaf-level SIF calculated by the $f_{esc\_GPR-SR}$ model is significantly smaller than that of multi-angle canopy SIF, indicating that the $f_{esc\_GPR-SR}$ model can eliminate the directional effect of the canopy SIF effectively. Figure 9 displays the comparison of $f_{esc}$ in the near-infrared band estimated by the NIRv/FAPAR model and the $f_{esc\_GPR-SR}$ model with that simulated by the DART model. The $f_{esc}$ estimated by the $f_{esc\_GPR-SR}$ model is closer to the 1:1 line, with similar $R^2$ values but the RMSE decreasing significantly from 0.070 to 0.026, in comparison with the NIRv/FAPAR model. These results of the 3-D radiation transfer model also verify that the $f_{esc\_GPR-SR}$ model can effectively improve the accuracy of $f_{esc}$ estimation.

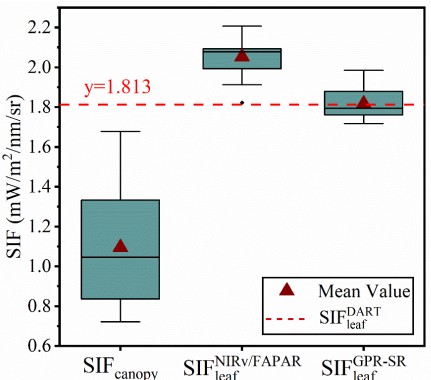

**Figure 8.** Boxplot of maize canopy SIF in the near-infrared band (760 nm) simulated by the DART model ($SIF_{canopy}$) and leaf-level SIF estimated by the NIRv/FAPAR model ($SIF_{leaf}^{NIRv/FAPAR}$) and that estimated by the $f_{esc\_GPR-SR}$ model ($SIF_{leaf}^{GPR-SR}$). The red dotted lines represent the leaf-level SIF of maize in the near-infrared band (760 nm) simulated by the DART model ($SIF_{leaf}^{DART}$). For the convenience of comparison, the unit of leaf-level SIF is converted to mW/m²/nm/sr (consistent with the unit of canopy SIF).

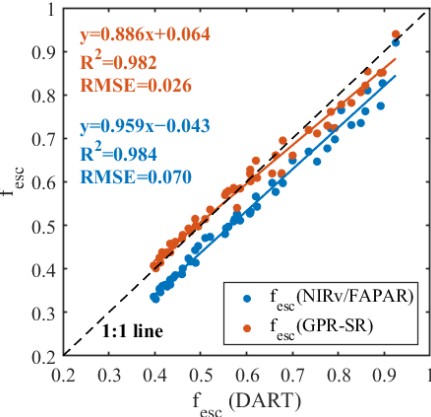

**Figure 9.** Comparison of $f_{esc}$ in the near-infrared band (760 nm) simulated by the DART model ($f_{esc}$ (DART)) with that estimated by the NIRv/FAPAR model ($f_{esc}$ (NIRv/FAPAR)) and the $f_{esc\_GPR-SR}$ model ($f_{esc}$ (GPR-SR)). $R^2$ is the correlation coefficient of linear regression, and RMSE is the root mean square error between $f_{esc}$ (DART) and the $f_{esc}$ values calculated by $f_{esc}$ estimation models.

### 3.3. Evaluation of the $f_{esc\_GPR-SR}$ Model Using Field-Measured Data

The $\Phi_{SIF}$ in Equation (1) remains relatively unchanged under high-light and stress-free conditions [49,50]. As a result, there is a strong correlation between $APAR_{green}$ and $SIF_{total}$, which is stronger than at the canopy level, taking into account canopy scattering and reabsorption [15,26]. We use the field-measured data from healthy and non-stressed vegetation to study the correlation of $APAR_{green}$–SIF at both canopy and leaf levels to assess the performance of both downscaling methods.

To verify the improved performance of the model that accounts for soil reflectance in SIF downscaling, the samples with FVC ≤ 0.8 from the ground-measured dataset were selected for comparison and validation. Figure 10 reveals the correlation between $APAR_{green}$ and the canopy-level SIF, the leaf-level SIF calculated by the NIRv/FAPAR model, and the $f_{esc\_GPR-SR}$ model for various species. In comparison with the $APAR_{green}$–$SIF_{canopy}$ relationship, the $R^2$ of the correlation between SIF after downscaling using both models and $APAR_{green}$ increases significantly and the slopes of the linear regression lines of $APAR_{green}$–$SIF_{leaf}$ for various species become more similar, indicating that downscaling SIF from canopy level to leaf level eliminates the impact of varying canopy structures among species and reduces the species dependence of the $APAR_{green}$–SIF correlation. Our $f_{esc\_GPR-SR}$ model performs better in this aspect, as reflected by the fact that the $R^2$ of the relationship between $APAR_{green}$ and $SIF_{leaf}$ estimated by the $f_{esc\_GPR-SR}$ model improves from 0.921 to 0.937 and the RMSE declines from 0.904 to 0.656 mW/m$^2$/nm in comparison with the results of the NIRv/FAPAR model.

The $R^2$ and RMSE values of the linear correlation between $APAR_{green}$ and SIF for samples of various species with FVC ≤ 0.8 appear in Table 6. The results indicate that the $R^2$ values of the $APAR_{green}$–$SIF_{leaf}$ relationship for different species increase after the SIF downscaling. Compared with the NIRv/FAPAR model, the RMSE of $SIF_{leaf}$ for different species calculated by our $f_{esc\_GPR-SR}$ model is smaller, indicating that the $f_{esc\_GPR-SR}$ model can better estimate the leaf-level SIF. For vegetables and crops, although the $R^2$ value slightly decreases after accounting for soil reflectance, the RMSE is still smaller than that of the NIRv/FAPAR model. For gold coin grass and winter wheat, the $SIF_{leaf}$ calculated by the $f_{esc\_GPR-SR}$ model displays a better linear correlation with $APAR_{green}$.

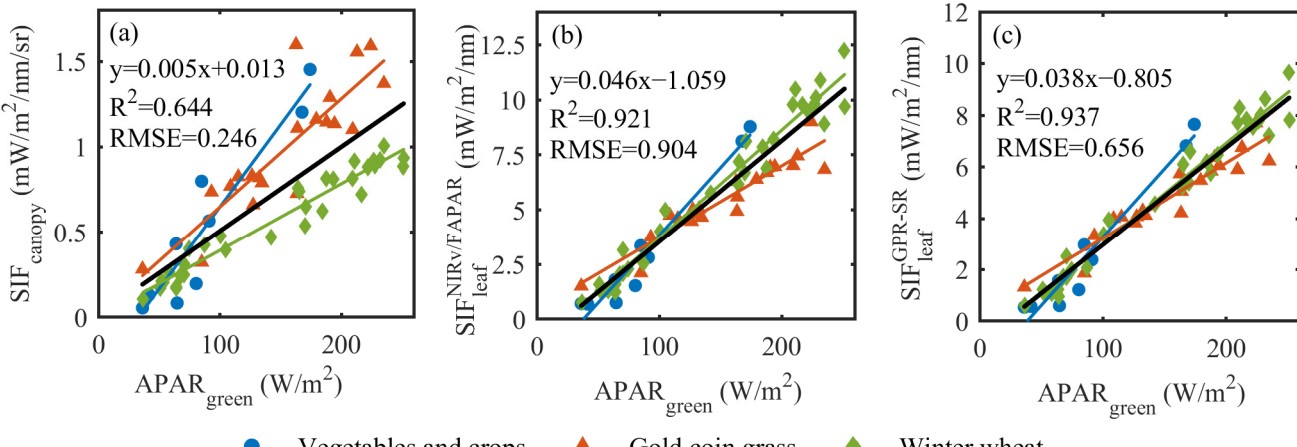

**Figure 10.** Correlation between $APAR_{green}$ and (**a**) canopy-level SIF ($SIF_{canopy}$), (**b**) leaf-level SIF estimated by the NIRv/FAPAR model ($SIF_{leaf}^{NIRv/FAPAR}$), and (**c**) leaf-level SIF estimated by the $f_{esc\_GPR-SR}$ model ($SIF_{leaf}^{GPR-SR}$) for different species (vegetables and crops, gold coin grass, and winter wheat) in the case of FVC $\leq$ 0.8. The black solid lines and the equations are the linear regression lines and models for all samples with FVC $\leq$ 0.8. The finer colored lines are the linear regression lines of the species that correspond in color.

**Table 6.** $R^2$ and RMSE values of the line correlation between $APAR_{green}$ and SIF for samples of various species with FVC $\leq$ 0.8. The units of RMSE for the $APAR_{green}$–$SIF_{canopy}$ relationship and the $APAR_{green}$–$SIF_{leaf}$ relationship are mW/m$^2$/nm/sr and mW/m$^2$/nm, respectively.

| | Vegetables and Crops | | Gold Coin Grass | | Winter Wheat | |
|---|---|---|---|---|---|---|
| | $R^2$ | RMSE | $R^2$ | RMSE | $R^2$ | RMSE |
| $SIF_{canopy}$ | 0.883 | 0.187 | 0.750 | 0.195 | 0.944 | 0.071 |
| $SIF_{leaf}^{NIRv/FAPAR}$ | 0.955 | 0.709 | 0.891 | 0.612 | 0.966 | 0.683 |
| $SIF_{leaf}^{GPR-SR}$ | 0.950 | 0.646 | 0.901 | 0.518 | 0.970 | 0.511 |

Overall, the outcomes indicate that the $f_{esc\_GPR-SR}$ model shows better performance in SIF downscaling for different species, reducing the species dependence of $APAR_{green}$–SIF for sparse vegetation to a greater extent in contrast to the NIRv/FAPAR model.

## 4. Discussion

### 4.1. Effect of Soil Reflectance on Estimating $f_{esc}$

The scattering and reabsorption of SIF photons within the canopy are governed by the same physical mechanisms as the scattering and reabsorption of reflected radiation photons. When incident light enters the canopy from the top, it can either pass through the canopy and reach the soil surface through the gaps or interact with leaves in the canopy. Photons intercepted by the canopy are scattered and reabsorbed several times, and some escape from the canopy while others are absorbed by leaves. Photons absorbed by leaves in the PAR range (400–700 nm) can stimulate fluorescence photons with a 640–850 nm wavelength range [27]. Emitted fluorescence will also escape from the canopy after multiple scatterings and reabsorptions inside the canopy and leaves.

In practice, soil background has a specific reflectance spectrum and is not "black." Ignoring the influence of atmospheric radiative transfer and multiple scatterings between soil and canopy (considering only single scattering), the photons captured by the sensor mainly come from three sources: (1) photons (including emitted fluorescence photons) that escape upward from the canopy to the sensor; (2) incident photons that pass through the canopy, reach the soil surface, and are reflected to the sensor; (3) photons (including emitted fluorescence photons) escaping downward from the canopy that reach the soil

and are reflected to the sensor. It is worth mentioning that the downward-escaping SIF photons from the canopy may be absorbed by the soil; however, this can be ignored as the lower leaves receive less light and thus produce fewer SIF photons [15]. Consequently, the SIF photons captured by the sensor come from two main sources: the contribution of pure vegetation canopy and the contribution of soil single scattering, and soil reflectance affects both. Yang et al. [27] proposed a basic model to estimate $f_{esc}$ based on spectrally invariant theory, where $f_{esc} = \text{Ref}_{NIR}/i_0 \cdot \omega_N$, ignoring the contribution of soil scattering to the fluorescence signal captured by the sensor. The near-infrared reflectance in the numerator clearly contains the effect of soil reflectance. Zeng et al. [28] proposed the NIRv/FAPAR model, which replaced the near-infrared reflectance with NIRv; however, the NDVI used as the pure vegetation signal in this model still depended on soil reflectance. The accuracy of $f_{esc}$ estimated by the NIRv/FAPAR model is reduced when a real soil background, rather than a non-reflecting background, is present, especially for sparse scenes [51]. This is because soil background pollutes the NIRv used to calculate $f_{esc}$. The variability of the value of $i_0$ is also high for sparse vegetation, which indicates that soil background can affect $i_0$ and thus $f_{esc}$ [15].

The simulations using the SCOPE model in this study also show that soil reflectance significantly affects the estimation of $f_{esc}$, particularly for sparse vegetation (Figure 11). When NIRv < 0.439, soil reflectance is the dominant factor in $f_{esc}$, influencing the scattering process between the canopy and soil background. The canopy structure is the dominant factor of $f_{esc}$ when NIRv > 0.439. Thus, ignoring the reflection characteristic of the soil background and treating it as "black" will introduce significant uncertainty in the calculation of $f_{esc}$, especially for sparse canopies where soil reflectance has a greater influence.

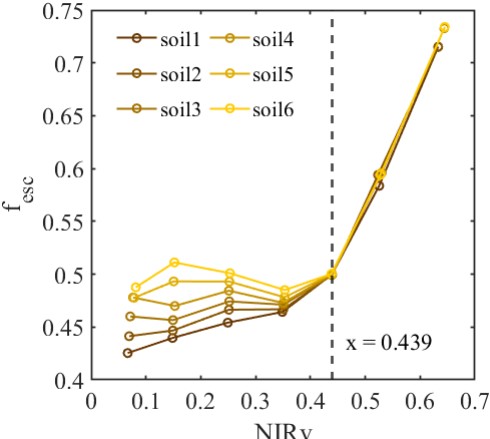

**Figure 11.** Relationship between $f_{esc}$ and NIRv simulated by the SCOPE model. The small circles in the figure represent the average values of different NIRv segments. The six curves in different colors represent the six soil spectral curves used in the SCOPE simulations.

### 4.2. Superiority of the $f_{esc\_GPR\text{-}SR}$ Model

At present, some studies on SIF downscaling have considered soil reflectance; however, these studies lacked clarity and had significant uncertainties. Liu et al. [15] proposed a SIF-downscaling approach where the ratio of $f_{esc}$ to the bi-directional reflectance factor was obtained through SCOPE model simulations and the hypothesis of "black soil" was ignored in the machine learning process. However, they did not explicitly account for the effect of soil reflectance. Zhang et al. [52] developed a method to derive the global soil-resistant $SIF_{total}$ ($SIF_{total\text{-}SR}$) using satellite data. However, their method used an approximation of the observed minimum reflectance to soil reflectance that is uncertain and relied on LAI and clumping index (CI) satellite data, which also propagated uncertainties in $SIF_{total}$. Notably, Zeng et al. [53] proposed using NIRvH with minimal soil impacts by making use of the spectral shape changes in the red-edge region to calculate true vegetation near-infrared

reflectance. We used 9000 validation samples simulated by the SCOPE model to verify the performance of the NIRvH2/FAPAR model to estimate $f_{esc}$ (Figure S4). Compared with the results in Figure 5, it can be found that our model is also superior to the NIRvH2/FAPAR model ($R^2$ = 0.885, RMSE = 0.051), possibly because of uncertainties in NIRvH itself, including the assumption that NIRvH2 is based on a linear increase in the reflectance of soil in the red-edge region which is not always correct.

Ma et al. [54] suggested that supervised machine learning methods trained on appropriate training datasets could construct accurate predictive models and overcome difficulties in physical modeling. Rasmussen et al. [55] also noted that while traditional parametric models are easy to interpret, they can be limited in their expression for complex datasets. Hence, in this study, we chose to build our $f_{esc\_GPR-SR}$ model using a machine learning method. We proposed to add a correction factor composed of soil reflectance and NDVI to the widely used NIRv/FAPAR model suggested by Zeng et al. [28], in order to explicitly account for the effect of soil reflectance. This not only preserved the advantages of the NIRv/FAPAR model's simplicity, ease of computation, and clear physical meaning but also compensated for its lack of explicit consideration of soil reflectance's effect on $f_{esc}$ estimation. We tested the performance of various machine learning algorithms and found that exponential GPR was the best training model. Directly using the machine learning model to estimate $f_{esc}$ may lead to poor model robustness due to too many model input parameters, so the correction factor $f$ ($Ref_{soil}$, $VI$) with fewer parameters was only trained using the exponential GPR method, which helped to reduce the number of input parameters and improve the model's robustness. Moreover, it ensured that the improvement was carried out on the basis of retaining the physical meaning of the original NIRv/FAPAR model as much as possible.

We demonstrated the superiority of our model by combining simulation data with field data. The validation results of SCOPE model simulations (Section 3.2.1) showed that the $f_{esc}$ estimated by the $f_{esc\_GPR-SR}$ model was in good agreement with the true $f_{esc}$ values, even under sparse vegetation, and it can significantly eliminate the influence of direction effect. In addition, the 3-D radiative transfer model dataset also validated that the $f_{esc\_GPR-SR}$ model could improve the estimation of $f_{esc}$ (Section 3.2.2). Since the DART model is too time-consuming to use for simulating a large amount of training data, a small dataset was used to evaluate the performance of the $f_{esc}$ estimation model trained with the SCOPE simulations. Additionally, ground-measured data from healthy and non-stressed vegetation were used for supplementary verification and showed that the linear correlation between $APAR_{green}$ and $SIF_{leaf}$ could be improved for different species when soil reflectance is considered and FVC is less than or equal to 0.8. The $f_{esc\_GPR-SR}$ model reduced the species dependency of the SIF–$APAR_{green}$ relationship for sparse vegetation to a greater extent (Section 3.3). The $R^2$ value of the $APAR_{green}$–$SIF_{leaf}$ correlation for vegetables and crops decreased slightly after considering soil reflectance, which may be because most of the photographs of vegetables and crops at Nanbin Farm were taken on cloudy days, leading to greater uncertainty in the $FAPAR_{green}$ and soil reflectance calculations using the photographs. It must be noted that the significant linear relationship between $APAR_{green}$ and SIF exists only in the absence of environmental stress. When environmental stress exists, $APAR_{green}$ may not be able to characterize the leaf-level SIF. So, $APAR_{green}$ cannot simply be used as a proxy of the total SIF. In conclusion, the $f_{esc\_GPR-SR}$ model, which accounts for soil reflectance, has been verified to increase the accuracy of $f_{esc}$ estimation in the near-infrared band, particularly for sparse vegetation and the leaf-level SIF in the near-infrared band calculated by the $f_{esc\_GPR-SR}$ model is less sensitive to observation angles and variations in canopy structure among multiple species.

### 4.3. Uncertainties of the $f_{esc\_GPR-SR}$ Model

Although the $f_{esc\_GPR-SR}$ model has been shown to be superior, there are still uncertainties in the modeling process. Firstly, in addition to the influence of soil background reflectance, the assumption that $\omega_N$ was 1 also contributed to the underestimation of $f_{esc}$

by the NIRv/FAPAR model, because $\omega_N$ is actually less than 1. However, considering that $\omega_N$ is the sum of leaf reflectance and transmittance in the near-infrared band and the absorption effect of leaves in this band is very weak, numerous studies have shown and acknowledged that $\omega_N$ is relatively stable and close to 1 [15,27–29]. Therefore, we did not pay attention to the uncertainty caused by the assumption that $\omega_N$ was 1 but focused on the effect of soil reflectance on $f_{esc}$ estimation. In the future, improving the precision of $f_{esc}$ can be considered by calculating $\omega_N$ accurately. Secondly, in this study, the correction factor was modeled using a combination of soil reflectance and the vegetation index NDVI, which represents vegetation coverage. While this combination was selected after evaluating the models' performance using four common vegetation indices, the correction factor could be further optimized in the future by testing more than two multi-factor combinations. Thirdly, despite the computational advantages of machine learning methods, they are still black-box models and heavily dependent on training datasets, which reduces their adaptability in special conditions. In the future, we plan to improve the model by establishing a new semi-empirical analytical model that considers the influence of soil reflectance on $f_{esc}$ estimation in two parts (one is the influence on the calculation of pure vegetation reflectance and the other is the influence of single scattering between soil and canopy), based on the basic model ($f_{esc} = Ref_{NIR}/(i_0 \cdot \omega_N)$) proposed by Yang et al. [27]. Additionally, there are still uncertainties that cannot be clearly analyzed and quantified, including the retrieval error of SIF. In a word, there are still many problems to be addressed in future SIF-downscaling research.

## 5. Conclusions

Accurate estimation of the canopy fluorescence escaping probability is important for SIF application, but the current algorithms cannot well deal with the influence of soil reflectance, especially for sparse vegetation. In this work, a correction factor estimated using the GPR algorithm with soil reflectance and NDVI was introduced into the widely used NIRv/FAPAR model for better estimation of $f_{esc}$ in the near-infrared band. The new method we proposed, the $f_{esc\_GPR-SR}$ model, was evaluated using simulation data and ground-measured data. The validation results of two simulation datasets from the SCOPE model and the DART model demonstrate that the performance of the $f_{esc\_GPR-SR}$ model in estimating $f_{esc}$ is significantly better than that of the NIRv/FAPAR model, particularly for sparse vegetation, and our $f_{esc\_GPR-SR}$ model can also effectively eliminate the influence of direction effects. Moreover, the validation results using the in situ measured data also prove that, compared with the NIRv/FAPAR model, our $f_{esc\_GPR-SR}$ model can better reduce the species dependence of $APAR_{green}$–SIF and eliminate the effect of canopy structure difference in multiple species. This study highlights the significance and advantages of considering soil reflectance in $f_{esc}$ modeling and presents a more accurate $f_{esc}$ estimation model, which will be useful for further studies on the SIF–GPP relationship.

**Supplementary Materials:** The following supporting information can be downloaded at: https://www.mdpi.com/article/10.3390/rs15184361/s1, Figure S1: Soil spectra called "loam_gravelly_brown_dark" used in the DART model (from the DART model database "Lambertian.db"); Figure S2: (a) Nadir and (b, c) side view of the 3-D canopy of maize simulated using the DART model. The size of the scene is 1.5 m × 1 m, and 20 maize are planted in two rows, with 10 in each row. Figure S3: Training performance (Predicted vs. Actual) of machine learning algorithms based on 5-fold cross-validation method when the input parameters are soil reflectance at 780 nm and NDVI; Figure S4: Comparison of $f_{esc}$ in the near-infrared band (760 nm) estimated by the NIRvH2/FAPAR model with the $f_{esc}$ simulated by the SCOPE model. $R^2$ is the correlation coefficient of linear regression, and RMSE is the root mean square error between $f_{esc}$ (SCOPE) and the $f_{esc}$ values calculated by the NIRvH2/FAPAR model; Table S1: The main parameters of the GPR model when using MATLAB to fit Gaussian process regression.

**Author Contributions:** Conceptualization, X.L. and L.L.; methodology, M.Q. and X.L.; software, M.Q.; formal analysis, M.Q.; investigation, M.Q.; resources, X.L., S.D. and L.L.; data curation, M.Q. and S.D.; writing—original draft preparation, M.Q.; writing—review and editing, X.L., L.L., L.G. and R.C.; visualization, M.Q.; funding acquisition, X.L. and L.L. All authors have read and agreed to the published version of the manuscript.

**Funding:** This research was funded by the National Key Research and Development Program of China, grant number 2022YFF1301900, and the National Natural Science Foundation of China, grant number 42071310.

**Data Availability Statement:** Data will be made available upon request.

**Conflicts of Interest:** The authors declare no conflict of interest.

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
