# Peer review of "Improving the Estimation of Canopy Fluorescence Escape Probability in the Near-Infrared Band by Accounting for Soil Reflectance"

_remotesensing, doi:10.3390/rs15184361_

Round 1
Reviewer 1 Report
General comments:
This manuscript proposed a method that considers soil albedo (reflectance?) to improve further the estimation of the escape probability of far-red SIF. Overall, very well done. This manuscript will be of interest to the broader SIF research community. However, some issues still need further clarification and revision, as detailed in the comments below.
Main concerns:
1. In this paper, the data you use is soil reflectance or soil albedo? Written at L137 is the soil reflectance at 780 nm.
2. The method in this article is for far-red SIF, which you need to point out in the article. There are also some expressions in the article that need to be unified, some places express far-red SIF (e.g., L56), and some places express near-infrared SIF (e.g., L97).
3. As shown in Figures 5, 6 and 9, fesc(NIRv/FAPAR) is smaller than the reference value for both SCOPE and DART simulations. This reason is not only the influence of the soil background, but also the assumption that wN is 1. In fact, wN is less than 1. And why can your method improve this problem? Should be discussed further in the Discussion section.
4. Why is APARgreen used as an evaluation criterion for your method? If APARgreen can be used as a standard, why spend so much effort to downscale to get the leaf-level SIF? This is a question worth considering.
Specific comments:
L12: stimulated -> emitted
L88: the photosynthetic active radiation absorbed by vegetation (FAPAR) -> FAPAR
Reviewer 2 Report
1. A brief description of the "black soil background" is needed in the Introduction
2. What is the effect of considering LAI, SZA, and other features retrieved from a satellite over the simulated input variables from the SCOPE?
3. Similarly, is it possible to use the actual satellite-retrieved data while validating the performance and comparing the validation with simulated values and actual retrievals?
4. Multiple remote sensing indices are mentioned in Table 2, but the sources, spatial and temporal resolution, and other details should be mentioned.
5. Table S1 not found
6. Machine learning model representation in equations 8 and 9 needs to be rewritten. The models are not clear
7. Justify why the Fesc_GPR_SA model performs well for all LAI values while the Fesc does not
8. Please write two or three sentences on what is meant by SIF downscaling in your research perspective
9. Conclusion can be improvised by giving more insights into the work
10. Most of the references are older than the last five years. Update the manuscript with the latest references.
Minor editing needed.
Round 2
Reviewer 1 Report
The authors answered my doubts well. The manuscript has been greatly improved.